# Regional Assessments of Surface Ice Elevations from Swath-Processed CryoSat-2 SARIn Data

**Natalia Havelund Andersen** [1,*], **Sebastian Bjerregaard Simonsen** [1], **Mai Winstrup** [1], **Johan Nilsson** [2] and **Louise Sandberg Sørensen** [1]

1    DTU Space, National Space Institute, Technical University of Denmark, 2800 Kgs. Lyngby, Denmark; ssim@space.dtu.dk (S.B.S.); maiwin@space.dtu.dk (M.W.); slss@space.dtu.dk (L.S.S.)
2    Jet Propulsion Laboratory, California Institute of Technology, Pasadena, CA 91109, USA; johan.nilsson@jpl.nasa.gov
*    Correspondence: naand@space.dtu.dk

**Abstract:** The Arctic responds rapidly to climate change, and the melting of land ice is a major contributor to the observed present-day sea-level rise. The coastal regions of these ice-covered areas are showing the most dramatic changes in the form of widespread thinning. Therefore, it is vital to improve the monitoring of these areas to help us better understand their contribution to present-day sea levels. In this study, we derive ice-surface elevations from the swath processing of CryoSat-2 SARIn data, and evaluate the results in several Arctic regions. In contrast to the conventional retracking of radar data, swath processing greatly enhances spatial coverage as it uses the majority of information in the radar waveform to create a swath of elevation measurements. However, detailed validation procedures for swath-processed data are important to assess the performance of the method. Therefore, a range of validation activities were carried out to evaluate the performance of the swath processor in four different regions in the Arctic. We assessed accuracy by investigating both intramission crossover elevation differences, and comparisons to independent elevation data. The validation data consisted of both air- and spaceborne laser altimetry, and airborne X-band radar data. There were varying elevation biases between CryoSat-2 and the validation datasets. The best agreement was found for CryoSat-2 and ICESat-2 over the Helheim region in June 2019. To test the stability of the swath processor, we applied two different coherence thresholds. The number of data points was increased by approximately 25% when decreasing the coherence threshold in the processor from 0.8 to 0.6. However, depending on the region, this came with the cost of an increase of 33–65% in standard deviation of the intramission differences. Our study highlights the importance of selecting an appropriate coherence threshold for the swath processor. Coherence threshold should be chosen on a case-specific basis depending on the need for enhanced spatial coverage or accuracy.

**Keywords:** swath processing; ice elevations; CS2; validation

## 1. Introduction

Satellite-radar altimetry data are key in documenting the most recent changes in Earth's cryosphere [1–9]. Due to the radars' large beam-limited footprint (∼13 km in diameter), conventional radar altimetry has difficulties in mapping regions with highly varying surface relief. These areas are mostly located in the marginal zones of the ice sheet, and characterize most smaller ice caps or glaciers [10,11], where the largest changes to date have taken place in the form of widespread ice loss. Thus, it is vital to develop improved monitoring capabilities for these areas to help us better understand the contribution of these regions to present-day sea levels. Since 2010, state-of-the-art radar altimeter SIRAL onboard CryoSat-2 (CS2) has been used to map these challenging regions with the unique interferometric SAR (SARIn) technique [1]. This technique allows for so-called swath processing, a method that uses the majority of the radar return waveform to generate elevations beyond the point of closest approach (POCA). Swath processing was developed

using CS2 data in 2013 by [12], and it takes advantage of the dual-antenna system onboard CS2 and provides the capability of pinpointing echolocations on the ground by means of the differential interferometric phase. This allows for the usage of the entire waveform to generate across-track elevation estimates. Such increased spatial coverage is needed in more topographically challenging areas within the SARIn mode mask [11–13]. The conventional method of POCA tends to track topographic highs such as ridges inside the footprint of the radar, leaving areas at lower elevations unmeasured [14–16]. This issue is improved when using swath data, which provide a more detailed and less biased representation of the topography of lower areas. Further, improved surface coverage reduces the need for the spatial interpolation or extrapolation of observed elevation changes by, e.g., hypsometric averaging or Kriging methods [17–19].

Previous studies showed the great potential of swath processing to map high-resolution elevation changes [3,11,13,20]. Some impressive results from swath processing include the small-scale subsidence of the Bárðarbunga caldera in Iceland. In that study, the deflation of a magma chamber resulted in a lowering of the ice surface [11]. Other examples include the heterogeneous and rapid ice loss over the challenging regions of the Patagonian ice fields [20], and the development of a multisurface swath retracker for the mountainous region of Karakoram [21]. These studies show the complexity of building a reliable and stable swath processor, and it likely requires the regional tuning of the processor for optimal performance. The ongoing improvements of swath-processed data can also be seen with each new baseline where, among other things, roll-angle bias estimations and phase unwrapping are improved [22,23].

Here, we investigate the accuracy of surface elevations derived from swath processing using the new Baseline D data product for CS2 over four different regions in the Arctic (Figure 1); the Austfonna ice cap in Svalbard, regions around the Petermann glacier (Northwest Greenland), the Nioghalvfjerdsfjorden glacier (Northeast Greenland), and the Helheim glacier (East Greenland). These regions were chosen because they had been mapped by other sensors. Hence, we have independent validation data to assess the accuracy of the swath-processed data. Validation data consisted of both airborne X-band radar and laser data, and satellite laser altimetry from ICESat-2. We also included intramission CS2 crossover analysis to evaluate the elevation differences between two crossing satellite tracks from the same period in time.

In this study, we investigate how the accuracy of derived elevations varies within the swath, as this dictates how much of the waveform can be trusted in swath processing. Specifically, our investigation is focused on the selection of an appropriate coherence threshold for the processor, and its effects on measurement accuracy and density. Previous studies showed great potential in using swath processing. Here, we investigate the added value of regionally choosing different coherence thresholds—an aspect that has previously not been explored in the literature.

Through such detailed analysis of the regional performance of the swath processor, we contribute to the overall goal of improving ground coverage over these fast-changing regions to minimize the possible underestimation of land-ice volume changes observed from CS2 [24].

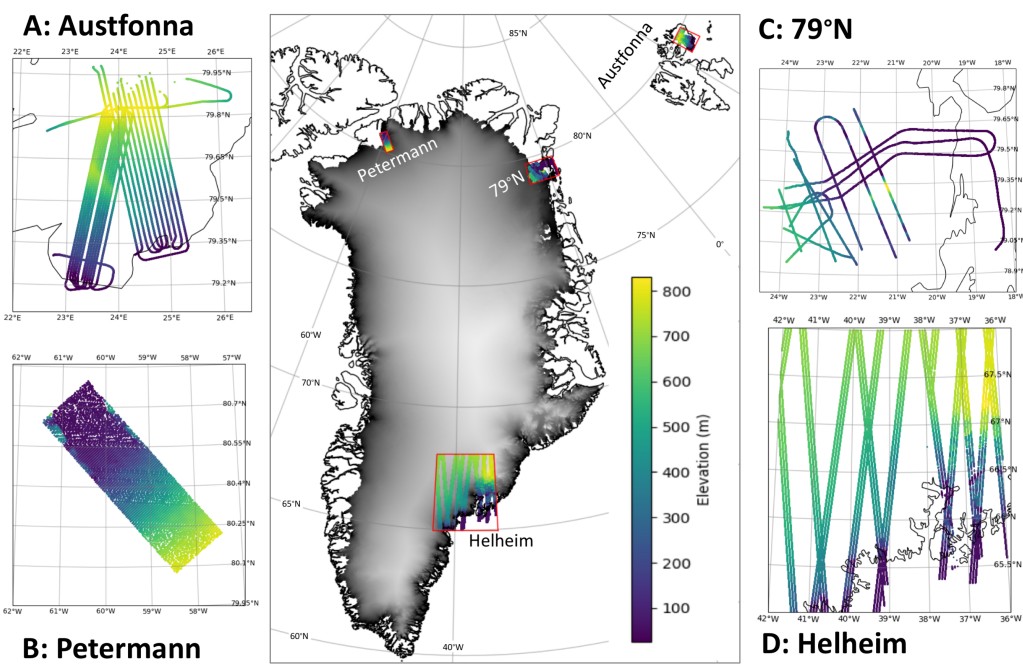

**Figure 1.** Location and coverage of four datasets used in this study. (**A**) Airborne ALS data from April 2016 over Austfonna ice cap in Svalbard; (**B**) airborne GeoSAR X-band data from April 2014 over Petermann glacier in Greenland; (**C**) operation Ice Bridge ATM data from 2017 over Nioghalvf-jerdsfjorden glacier; (**D**) ICESat-2 laser altimetry data for Helheim glacier obtained in June 2019. Gray-shaded data plotted as background on Greenland are 1 km resolution DEM from ArctiDEM [25].

## 2. Data and Regions of Interest

We computed swath-processed ice-surface elevations from SARIn data from CS2 in four Arctic regions (Figure 1 and Table 1). These regions were selected due to the availability of suitable independent high-resolution validation data.

**Table 1.** Overview of four validation datasets.

|   | Study Region | Period | Instrument |
|---|---|---|---|
| **A** | Austfonna ice cap | 1–30 April 2016 | ALS |
| **B** | Petermann glacier | 1–30 April 2014 | X-band Radar |
| **C** | Nioghalvfjerdsfjorden glacier | 1–30 April 2018 | OIB ATM |
| **D** | Helheim glacier | 9–24 June 2019 | ICESat-2 |

### 2.1. CS2 Data

European Space Agency (ESA) satellite CS2 was launched on 8 April 2010 with the main objective to measure changes in sea and land ice. The Ku-band Synthetic Aperture Interferometric Radar Altimeter (SIRAL) onboard the CS2 satellite operates in three different modes [26]: low-resolution mode (LRM), synthetic aperture radar (SAR), and SAR Interferometric (SARIn), all of which measure the surface elevation along the track every ∼300 m [1]. Only SARin delivers the differential phase required for swath processing. The SARIn mode is available in areas of highly complex and steep terrain, such as the margin of the ice sheets and over smaller ice caps. Three star-trackers onboard CS2 are used to determine the satellite altitude, orientation, and roll to allow for accurate echolocation mapping on the ground. In this study, Level 1b Baseline D data for SARIn mode are used.

For each waveform, the received power, phase, and coherence, along with the roll angle of the satellite, were available.

### 2.2. Airborne-Laser-Scanner Data over Austfonna Ice Cap

Austfonna is the largest ice cap in Svalbard and is located in the northeastern part of the archipelago [27]. On 15–16 April 2016, an airborne campaign mapped the surface topography of a large part of the Austfonna ice cap using a near-infrared airborne laser scanner (ALS). These measurements were collected as part of a Cryosat validation experiment (CryoVEx) campaign funded by ESA and carried out by DTU Space [28]. The data were used in studies to validate both CS2 SARIn Baseline C [29] and D [30] data. Since the airborne campaign was carried out to support the validation of CS2, the mapped grid was aligned with the actual ground tracks of the satellite. The resolution of the original ALS data is 1 m, and the scan is approximately 300 m wide. Here, we use an interpolated grid of 100 m resolution in order to mimic the footprint size of CS2. The area covered by the validation flight has highly variable topography, which includes a surging region characterized by crevasses. The coverage of the Austfonna ALS dataset is shown in Figure 1A.

### 2.3. Airborne X-Band Radar Data over Petermann Glacier

The Petermann glacier in Northwest Greenland is a major outlet glacier of the Greenland ice sheet (GrIS), and it has the second-largest floating ice tongue in Greenland. The surface topography of the Petermann glacier was mapped by GeoSAR on 4 April 2014. GeoSAR is a unique airborne dual-band, dual-sided interferometric radar mapping technology developed in a collaboration between NASA's Jet Propulsion Laboratory and Fugro EarthData. The GeoSAR is flown on a Gulfstream II jet, and the instrumentation collects data at X and P-band frequencies, which produces large differences in radar signal penetration [31]. In this study, we used X-band data. GeoSAR simultaneously measures the surface on both sides of the aircraft to generate high-quality digital elevation models. Swath width is approximately 12 km, pixel size is 1.25–3 m, and the absolute X-band DEM height error is less than 1 m [31]. The coverage of the Petermann X-band dataset is shown in Figure 1B. Data collection consisted of six primary mapping lines and one crosstie line. The coverage of the airborne X-band radar data over the Petermann glacier is shown in Figure 1B. Coinciding with the X-band mapping of Petermann, DTU also conducted an airborne laser campaign (ALS) of the area in connection with ESA CryoVEX 2014, resulting in a few available tracks over the glacier.

### 2.4. Operation Icebridge Airborne Topographic Mapper (ATM) Data over Nioghalvfjerdsfjorden Glacier

The Nioghalvfjerdsfjorden glacier (79 North Glacier) terminates in the largest ice shelf in Greenland. It drains the Northeast Greenland ice stream with the Zachariae and Storstrømmen outlet glaciers [32]. The Nioghalvfjerdsfjorden glacier was mapped by Operation IceBridge on 22 and 28 March, and 3 April 2017. Operation Icebridge carries a suite of instruments; here, we use surface elevations measured by the Airborne Topographic Mapper (ATM) provided in the IceBridge ATM L2 Icessn Elevation, Slope, and Roughness V002 data (ILATM2) product [33]. ATM surface elevations typically have a resolution of approximately 30 m along the flight track, varying with aircraft speed. The crossflight track scan width is represented by platelets of approximately 80 m. The coverage of the Nioghalvfjerdsfjorden ATM dataset is shown in Figure 1C.

### 2.5. ICESat-2 Laser Data over Helheim Glacier

The Helheim glacier is a large marine-terminating glacier in Southeast Greenland. The glacier has undergone significant changes in the past few decades, such as rapid dynamic thinning due to increases in ice-flow velocity between 2002 and 2005 [34]. Here, we use satellite laser altimetry from June 2019 over the Helheim region collected by the Advanced Topographic Laser Altimeter System (ATLAS) instrument onboard ICESat-2. ATLAS emits short pulses of laser light (532 nm) and uses a single-photon-counting detector recording

their travel times. ATLAS transmits pulses at 10 kHz, resulting in approximately one pulse every 0.7 m along the track. ATLAS transmits in six beams arranged in three pairs, each consisting of a strong and a weak beam. Each of these pairs is separated by about 90 m, and the beams are separated by 3 km. The ICESat-2 (IS2) footprint is less than 17 m horizontally.

Here, we use the downsampled ATL06 version 3 dataset (Land Ice Along-Track Height product) with a resolution of 20 m along the track, which is created from a ground-finding algorithm applied to individual photon detections in full-resolution ATL03 data [35]. The coverage of the Helheim region ICESat-2 dataset in June 2019 is shown in Figure 1D.

## 3. Swath Processing

Ice-surface elevations observed by radar altimeters are conventionally computed by tracking the POCA in the radar return waveform [12]. This is the only information that can be derived from radar altimeters operating in conventional pulse-limited configuration. The dual-antennas and the SIRAL instrument onboard CryoSat-2 allow for interferometric information to be included in the computation of the elevation and its location. This method is referred to as swath processing [10] since it enables a "swath" of elevation estimates to be generated from each waveform. Swath processing allows for geocoding surface echoes beyond the conventional POCA point, and can thus generate surface-elevation estimates in a wide swath, much like observations from scanning laser altimeters. Here, we provide a general overview of our swath-processing chain, which consists of three main steps. For the basic principles behind swath processing, we refer to Hawley et al. [10], Gourmelen et al. [11], Gray et al. [12].

Coherence threshold: The first step is to identify which parts of the radar return waveform are reliable to use for swath processing. This is performed by applying thresholds to remove parts of the waveform that are not strongly coherent or that are associated with low return power.

If the coherence of the data is high (high confidence that the signal recorded by the two antennas originated from the same point on ground), the coherence value approaches 1; if it approaches 0, it is dominated by noise. Requiring high coherence by setting a high coherence limit (e.g., 0.8) ensures that high-quality data are produced, but this limits the spatial data coverage. A more relaxed coherence threshold (e.g., 0.6) results in more elevation retrievals, but of lower quality. Areas with smooth and homogeneous topography tend to have overall high coherence and can yield extended coverage even with a high coherence threshold. Figure 2a shows an example of a waveform from a smooth part of the Austfonna ice cap from 26 April 2016, revealing a return signal with overall high coherence. Figure 2b shows the return signal from another area (also Austfonna, same data), with highly varying topography and large variability in coherence. Therefore, depending on the underlying topography, some areas can benefit from applying a lower coherence threshold in order to obtain better coverage, assuming the resulting elevations can be properly validated.

Filtering and phase unwrapping: An essential cornerstone of swath processing is to unwrap the phase information, i.e., correcting for local phase jumps of $\pm 2\pi$. However, differential phase $\delta\phi$ must first be filtered to ensure correct phase unwrapping. This is performed to avoid misplaced echoes in the final elevation product, as changes in the phase generate a direct change in the location of the radar echo. Filtering is achieved by creating a complex interferogram and then separately filtering with a wavelet denoising method of the real and imaginary phases [12,36]. The complex interferogram is then recombined to create the filtered differential phase. An example of phase unwrapping is shown in Figure 3, where the original phase in gray shows a phase jump between range bin 400 and 500. The red line shows the correctly unwrapped phase that is used for further processing. Prior to unwrapping, the low-coherence part of the phase is masked out to avoid the unwrapping of the high-coherence part of the signal being affected by the inherent noise in the low-coherence parts.

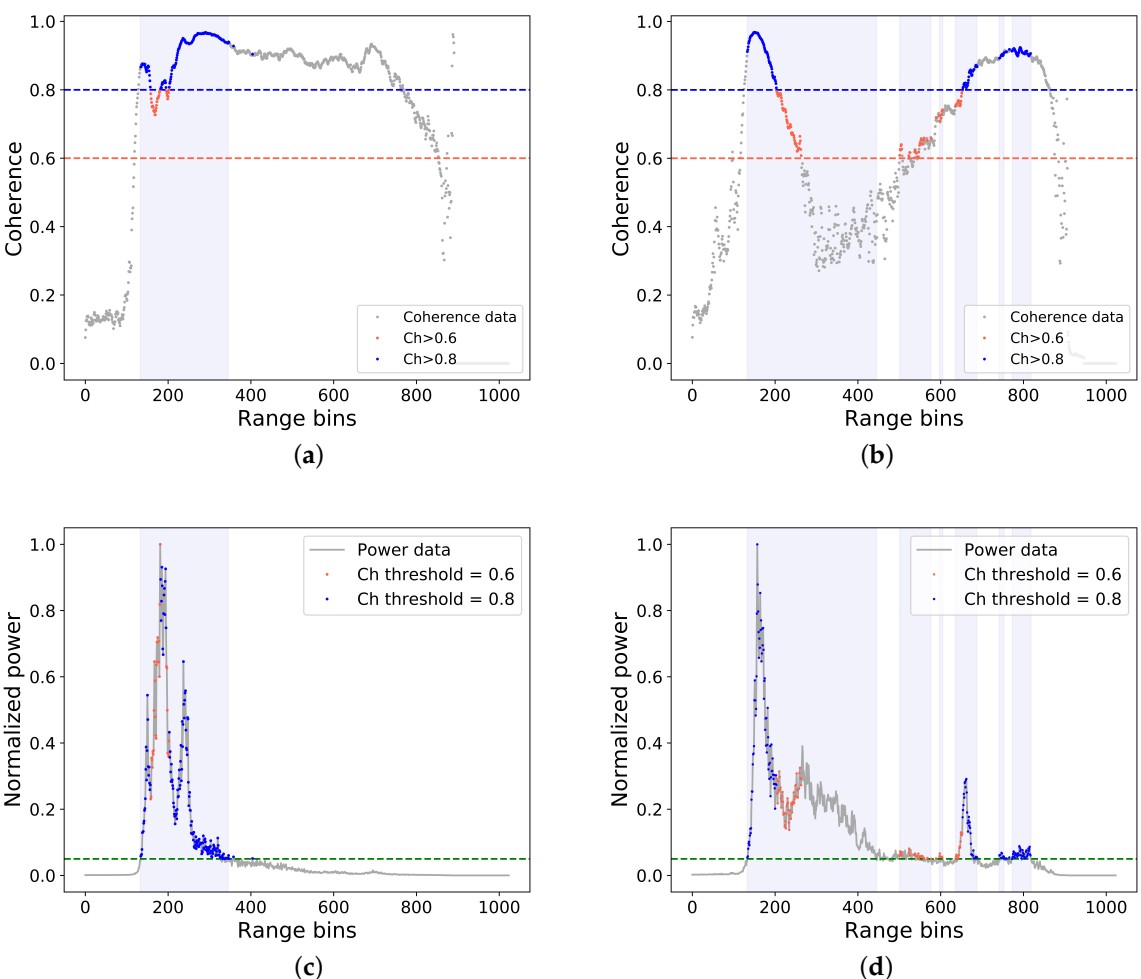

**Figure 2.** Coherence and power data from a CS2 track over Austfonna ice cap on 26 April 2016. (**a**,**c**) Northern part of the ice cap; (**b**,**d**) southern part of the ice cap. (**a**,**b**) Filtered coherence with threshold of 0.6 and 0.8 indicated by horizontal dashed lines; blue vertical box indicates area above the power threshold. (**c**,**d**) Power waveform, power threshold indicated by green line. Range bins for which the power was above the chosen limit are marked by shaded blue boxes.

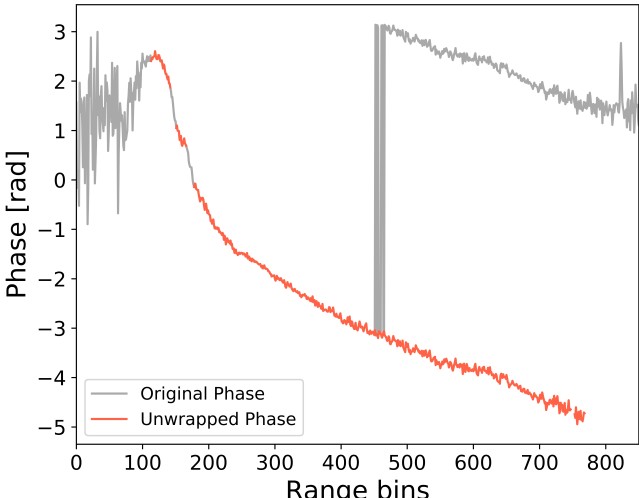

**Figure 3.** Example of local phase unwrapping: gray line, original wrapped phase; red line, position of correctly unwrapped phase for range bins with coherence above 0.8.

Geocoding and correcting global phase ambiguities: The elevation and associated geolocation is computed from the phase information and range-bin location by first deriving look angle $\theta$ from the differential phase [12] with

$$\theta = \arcsin \frac{\delta\phi}{k \cdot B} - \beta \tag{1}$$

where $\delta\phi$ is the differential phase; $k$ is the wavenumber; $B$ is the baseline length of the antenna; and $\beta$ is the satellite roll angle, which is included in the new Baseline D data [30]. Further, range to surface $R$ for the chosen bins along the waveform is given by [26]:

$$R = \frac{1}{2} \cdot c \cdot W_{delay} + R_b(Bin - Bin_{offset}) + corr \tag{2}$$

where $c$ is the speed of light; $W_{delay}$ is the window delay; $R_b$ is the range bin size (0.2342 m for SARIn); $Bin$ is the bin number used for this computation; $Bin_{offset}$ is the range bin offset (512 for SARIn); and $corr$ is the applied tidal and atmospheric propagation corrections from the Baseline D product [23,26]. Given the look angle and derived range, echo elevation $H$ can now be computed as

$$H = A - R \cdot \cos(\theta) + \frac{(R \cdot \sin(\theta))^2}{2 \cdot \rho} \tag{3}$$

where $A$ is the altitude of the satellite, and $\rho$ is Earth's curvature parameter to compute the echo elevation [37] (Equation (2.61)). The final geocoding ($lat$, $lon$) of the echo is then

$$lat = lat_{nadir} + \frac{R \cdot \sin(\theta) \cdot \cos(Az + \frac{1}{2}\pi)}{\rho_{lat}} \tag{4}$$

$$lon = lon_{nadir} + \frac{R \cdot \sin(\theta) \cdot \sin(Az + \frac{1}{2}\pi)}{\rho_{lon}} \tag{5}$$

where $lat/lon_{nadir}$ is the components of the nadir satellite position, $Az$ is the north azimuth, and $\rho_{lat/lon}$ is Earth's curvature parameter to compute the echo direction [37] (Equations (2.45) and (2.46)).

Even after phase unwrapping, phase ambiguities can still be present in the geocoded data, as the entire waveform can be shifted multiples of $2\pi$. This can be identified by comparing against an external reference DEM [12,36]. In the final step, we therefore investigated the difference between each waveform elevations and a reference DEM. In this study, the used reference DEM is release 7 of the ArcticDEM [38]. If the median difference was above 15 m, we computed a new set of geolocations by applying 2 multiples of $2\pi$ to the differential phase. The final set of swath heights and corresponding geolocations were chosen from the set of multiples of $2\pi$, which resulted in the lowest residuals [11,12,36]. Lastly, an ice mask from the Randolph Glacier Inventory was applied to ensure that the echo originated from an ice-covered area [39].

By applying the above steps, the swath processor could track the topography beyond POCA and reveal previously untracked topographic lows [11]. The improved method using swath processing can generate surface elevations from topographically challenging areas, and improve the coverage of small ice caps and the GrIS.

## 4. Validation Methods

To assess the reliability and performance of the swath processor, we relied on two metrics in validation analysis: (1) CS2 intramission crossover elevation differences and (2) external-mission crossover elevation differences.

The intramission CS2 crossovers of swath data were used to assess relative precision by evaluating the elevation differences between crossing ascending and descending satellite

passes within a given month. The 1 month temporal limit was chosen to reduce the physical-elevation change between ascending and descending satellite passes (due to, e.g., weather or ice dynamics), while preserving enough valid comparison samples. In previous CS2 data releases (prior to Baseline D), the satellite roll angle manifested itself as intramission biases in the CryoSat-2 Level 1b data, but this was greatly improved with the current Baseline D [22,30].

External-mission validation is performed assessing the differences between CS2 swath elevations and an external validation dataset. Here, only data located closer than 50 m in space and less than 31 days in time were compared. Data used for this external-crossover validation consisted of both laser and radar data (see Section 2). Due to the different nature of these sensors, we expected an offset between the swath elevations and some of the validation datasets from differences in snow penetration at the Ku band compared to the X band.

The difference in statistics between swath-processed elevations and the validation datasets was computed for all four areas using two different coherence limits (0.6 and 0.8). This allowed for investigating the potentially added value in increased coverage due to a relaxed coherence limit.

## 5. Results

The results of the intramission crossover statistics for the four regions and the comparison against the independent validation datasets are shown in Table 2, showing the number of elevation differences used in analysis, their median, and their standard deviation when applying 0.6 and 0.8 coherence thresholds, respectively. The resulting increase in the standard deviation of the elevation differences when decreasing the coherence limit from 0.8 to 0.6 is given in percentage in the parentheses.

**Table 2.** Overview of results from the crossover analysis on swath-processed surface-elevation differences. All results were corrected for an ascending or descending bias for the CS2 data of 30 cm. For each area, the number of crossover data points used in the analysis, and the median and standard deviation of the elevation differences are given. The increase in standard deviation from decreasing the coherence threshold is given in parentheses.

| | Area | Coherence Threshold | Crossover Points | Median [m] | Standard Deviation | Instrument |
|---|---|---|---|---|---|---|
| Intra-mission | Petermann | 0.8<br>0.6 | 41,712<br>66,104 | −0.003<br>0.03 | 7.12<br>9.50 (33.4%) | CS2 |
| | Helheim | 0.8<br>0.6 | 13,140<br>28,138 | 0.07<br>0.03 | 7.18<br>10.84 (50.9%) | CS2 |
| | 79° N | 0.8<br>0.6 | 21,659<br>39,002 | −0.03<br>−0.02 | 10.46<br>14.01 (33.9%) | CS2 |
| | Austfonna | 0.8<br>0.6 | 15,703<br>25,139 | −0.06<br>0.04 | 8.63<br>13.78 (64.8%) | CS2 |
| External-mission | Petermann | 0.8<br>0.6 | 270,612<br>178,432 | 1.66<br>1.61 | 6.35<br>8.56 (34.8%) | X-band |
| | Helheim | 0.8<br>0.6 | 9397<br>15,521 | −0.15<br>−0.3 | 7.55<br>10.46 (38.5%) | IS2 |
| | 79° N | 0.8<br>0.6 | 4171<br>6376 | −1.19<br>−1.13 | 10.41<br>17.42 (67.3%) | ALS |
| | Austfonna | 0.8<br>0.6 | 3800<br>5573 | −1.44<br>−1.48 | 8.28<br>10.83 (30.7%) | ALS |

Our initial intramission analysis revealed an elevation bias of 30 cm between ascending and descending satellite passes. The value of this ascending or descending bias in the intramission crossover differences was derived from the average bias between the

four regions. The bias is believed to originate from Level1b data, and is therefore not a result from the swath-processed elevation procedure. Hence, it was removed prior to crossover analysis. Results in Table 2 were corrected for this bias to better reflect the actual performance of the swath processor, and with regard to the comparison with the external datasets. However, our conclusions would be the same if we had chosen not to correct for this bias.

The total number of data points generated by the swath processor within each region depends on the chosen coherence threshold. On average, the reduction in data points was 25% when increasing the coherence threshold from 0.6 to 0.8, but with large variability depending on the terrain. Although the standard deviation of the elevation differences was smaller in all regions when increasing the coherence threshold, the impact on the statistics was quite different for the four regions (see Table 2). For the Petermann and Nioghalvfjerdsfjorden regions, the intramission standard deviation increased by only 33% and 34%, respectively, when lowering the coherence threshold, while the bias remained low. However, spatial coverage significantly increased, as we can see in Figure 4 for Nioghalvfjerdsfjorden when relaxing the coherence threshold. In comparison, the improvement in standard deviation for the challenging region of Austfonna was in the order of 65%.

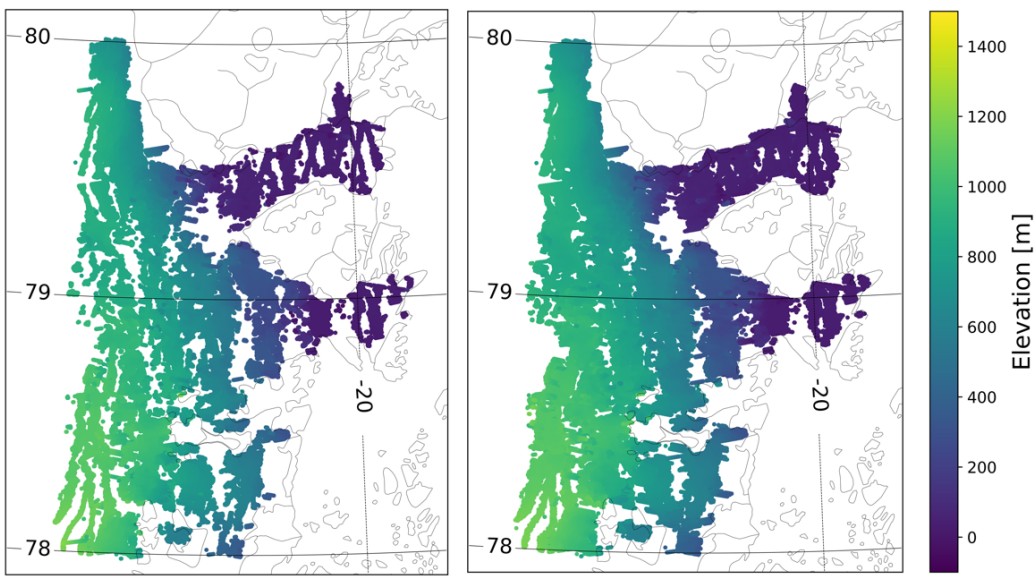

**Figure 4.** Swath-processed ice-surface elevations over the Nioghalvfjerdsfjorden glacier in April 2018 computed using the two different coherence thresholds. (**left**) Elevations computed using a coherence limit of 0.8, revealing gaps in the data. (**right**) Ice-surface elevations computed using coherence limit of 0.6, which left fewer gaps in the data.

Table 2 also shows that the median bias of the elevation differences was much less sensitive to the choice of coherence threshold. The median of the intramission elevation differences was found to be consistently below 7 cm for all four regions. However, it varied regionally in external mission analysis due to the regional difference in surface penetration and topographic relief.

The best agreement between CS2 and the external validation data was in the Helheim region, where CS2 was validated against IS2. Here, we found a bias of −0.3 m and a standard deviation of 10.46 m for a coherence threshold of 0.6. The spatial distribution of the crossover differences for the Helheim region is shown in Figure 5. The intra-ission elevation differences are shown in Figure 5a,b for coherence thresholds of 0.6 and 0.8, respectively. Figure 5c,d show the external-mission elevation differences for the same region. The largest bias of 1.66 m, in the Petermann region, was for the X-band validation data. The largest standard deviation of 17.42 m was found at the Nioghalvfjerdsfjorden glacier.

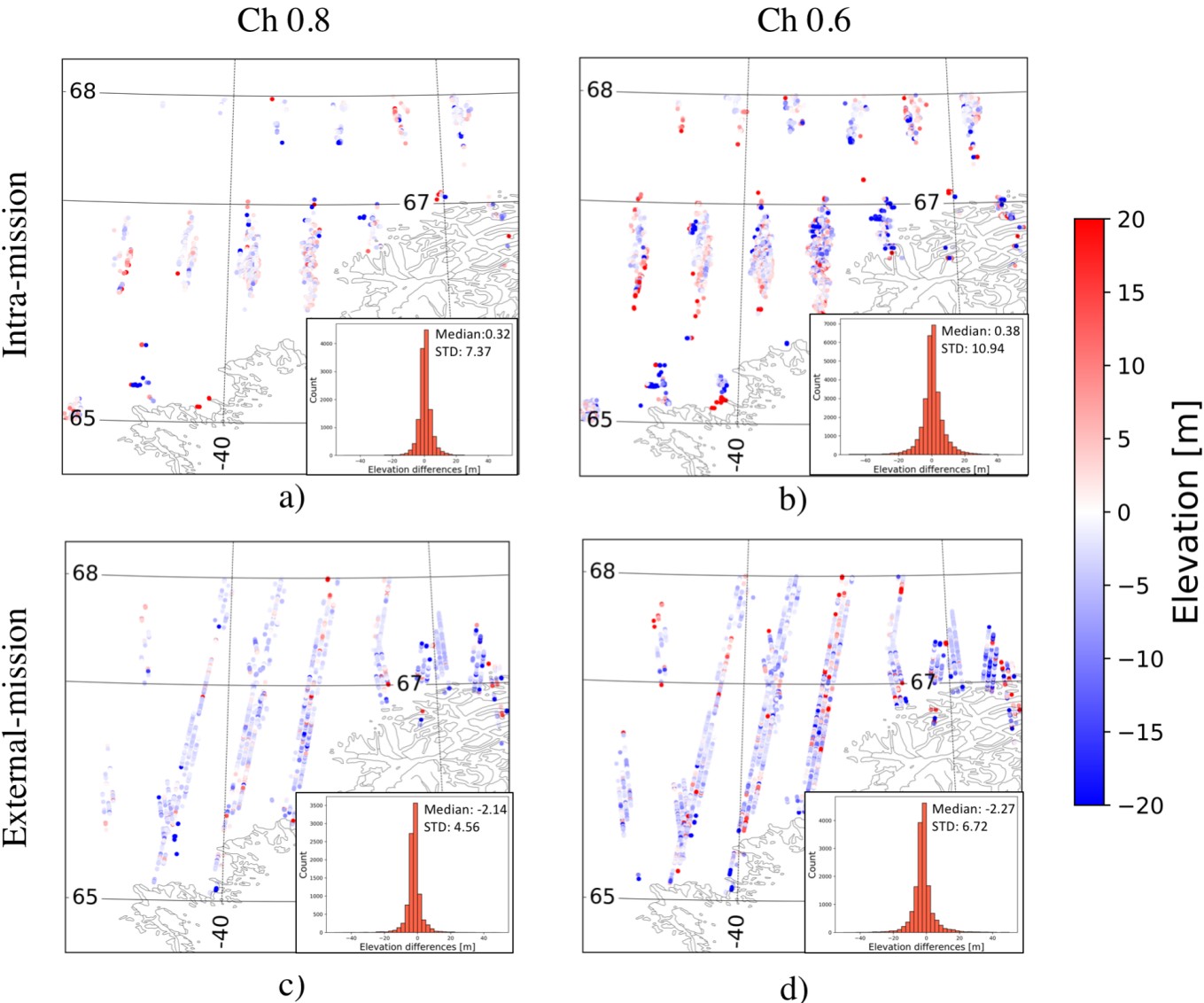

**Figure 5.** Results for Helheim area. (**a,b**) Intramission crossover elevation differences for coherence limits 0.8 and 0.6, respectively. (**c,d**) External-mission crossover analysis with ICESat-2 data for coherence limits 0.8 and 0.6, respectively. Corresponding histograms of elevation differences shown in inserts.

## 6. Discussion

An inherent result of relaxing the coherence limit is an increase in the number of surface elevation of an average of 25% for a coherence threshold of 0.6 versus that of 0.8. This increase in observations comes with the cost of an increased standard deviation of 33–65% and 31–67% for the intra- and external-mission elevation differences, respectively. Hence, the cost–benefit of relaxing the coherence limit is highly regional.

A bias of −1.48 m for the validation against the external dataset was found for the Austfonna ice cap. This ice cap has a complex topography [2], with a large number of surface crevasses as result of the 2012 surge of Basin 3 [40]. Therefore, it is a challenging region for radar altimetry; it was also noted to be a challenging area in the study by Sørensen et al. [29]. The 2012 surge of Basin 3 altered the surface slope in such a complex way that the employed reference DEM does not necessarily accurately represent the current surface. The external DEM was from 2018 and the data over Austfonna from 2014; hence, this is the largest temporal difference in this study. As a consequence, some errors could be introduced in the identification of global phase ambiguities. Some global phase ambiguities

may not be detected and still be present in the final result. On the other hand, as the topography changes, the DEM can also locate phase ambiguities that are not actually real, thereby introducing errors into the final result. The area most affected by the surge clearly affects the coherence of radar waveforms, as can be seen in Figure 2b, where a dip in coherence can be seen at around range bin 400. The generally low coherence in this area means that, in the "leading edge" of the waveform, most of the data can be used, while in the "trailing edge", a large part of the signal is deemed incoherent. The northern part of Austfonna, not affected by the surge, had a more uniform surface, with a slope ideal for swath processing. This resulted in high coherence in the majority of each waveform, as seen in Figure 2, allowing for us to use a larger part of the waveform. Hence, the effect of changing the coherence threshold may vary even between areas within the same region. This supports our hypothesis that it would be beneficial before swath processing to perform a regional assessment of the level of accuracy required for individual studies, as this is not a global constant.

The bias of the external-mission crossovers in the Petermann region was negative (i.e., validation data generally indicated slightly higher elevations than those in the swath elevation data); for the remaining regions, we found a positive bias. This was likely due to the different frequencies used by the instruments leading to differences in signal penetration depth. Results suggest that the penetration depth of the X-band instrument was greater than that of CS2, while smaller than those of IS2 and ALS. This agrees with our expectations based on the instruments' frequencies; the lower the frequency is, the greater the penetration depth, with the X band having the lowest frequency of the used instruments. At the Petermann glacier, we saw a clear difference in the median bias of the external-mission analysis when using X-band radar data as validation compared to laser. The swath elevations over the Petermann region showed the highest external-mission bias of 1.66 m for a coherence threshold of 0.8.

We computed the elevation difference between two different external datasets over the Petermann region to investigate the difference in snow-depth penetration depending on instrument frequency. Both ALS data (a few tracks) from CryoVex and X-band radar data over the Petermann glacier have been available from April 2014, and analysis of the elevation differences between these two airborne datasets revealed an average difference of 2.5 m. This bias could be directly attributed to surface penetration of the radar signal into the upper layers of the snowpack.

Helheim was the region with the best external-mission agreement for both coherence thresholds. The topography in the Helheim region is less complex than that of other places, leading to an average higher coherence for the waveforms. The median in this area was between −0.15 and −0.3 m for the coherence thresholds of 0.8 and 0.6, respectively. This bias was in the same range, as found between CS2 and GNSS stations in Antarctica [41]. However, much lower standard deviation was found between swath elevations and IS2. This shows that we can increase the spatial coverage by relaxing the coherence threshold in some areas without a large increase in error. The acquisition period for the validation dataset for the Helheim region was June 2019. During this time, there was very little snow cover in this region, which led to a more consistent scattering horizon between radar and laser data. The minimal snow cover was likely the reason for the excellent agreement observed in the external-mission analysis of the Helheim region. However, there was a decrease in radar penetration depth, which could have been caused by fluctuations in density due to summer melt events [42]. Furthermore, for Helheim, we analyzed the crossover differences to see if there was dependence on surface slope, and found that the standard deviation increased only slightly with increasing surface slope. Therefore, removing data points with large surface slopes (above 1.5 degrees) would not significantly impact the statistics of the crossover analysis.

Regional differences in our validation results underline the importance of regional tuning of the swath processor. Our study highlights that one should also consider using intramission differences rather than only comparing against an external dataset that might

not be optimal. The statistics of the external-mission elevation differences also reflect differences in, e.g., the type, sampling, and noise of the different instruments. Intramission elevation differences are not affected by these effects, and thereby to a large degree reflect measurement precision and repeatability.

Here, comparing radar-derived elevations with either the X band or the laser validation dataset impacts the result with an instrument bias component to crossover analysis. Airborne-laser-scanner data are valuable to assess, e.g., penetration depth, but a perfect match between CS2 and airborne-laser-scanner validation data that are most often acquired during springtime cannot be expected without considering the influence of penetration depth on snow. Since IS2 data are acquired throughout the year, summer data from here may be a better external validation set. In this study, IS2 measurements revealed the best external mission agreements, and thus seem better for the purpose of validation. This was likely due to the minimal snow cover in the summer months over the Helheim region. Seasonal variation in snow cover means that the agreement between CS2 and IS2 data may be less pronounced during the winter months, where snow results in a different scattering surface for the radar than that for the laser.

## 7. Conclusions

Swath elevation data from four different Arctic regions (Austfonna ice cap, Petermann glacier, Nioghalvfjerdsfjorden glacier, and Helheim area) were computed from CryoSat-2 SARIn Baseline D data, and validated through crossover statistics using temporally overlapping independent datasets and intramission crossovers. These validation datasets consist of both radar and laser data, namely, ALS data from CryoVEX, airborne X-band radar, Operation IceBridge, and ICESat-2 (ATL06) data. We investigated the results of crossover statistics by varying the coherence threshold in the swath processor and found that its effect varied between regions. In the Helheim region, the coherence threshold could be relaxed with only little effect on the bias of the intramission crossover differences. We recommend to use a relaxed coherence limit, as this increases the amount of data along the swath. For the Nioghalvfjerdsfjorden glacier, statistics significantly deteriorated, and the user should carefully consider if the increased data coverage is actually useful considering their larger uncertainties. On the basis of intramission crossovers, the largest increase in the standard deviation of the elevation differences (65%) when decreasing the coherence threshold from 0.8 to 0.6 was observed on the Austfonna ice cap. These regional differences in the statistics support our hypothesis that regional tuning contributes to attaining the optimal performance of the swath processor.

The choice of validation dataset impacted crossover differences in the external-mission comparison. An apparent external-mission bias was documented in those regions where X-band and airborne laser scanner data were used as validation. This was likely primarily caused by the difference in penetration depth among various instruments, which was also suggested by the presence of a bias of 2.5 m between laser and radar airborne data observed over the Petermann glacier region in April 2014. This emphasizes the significance of including intramission crossover differences for validation purposes and tuning the processor.

Furthermore, seasonality must be considered when investigating the agreement between laser and radar. The good agreement seen at Helheim in June would likely deteriorated during winter months due to an increase in snow cover and snow depth. The season is also a likely reason why the ALS from April did not agree completely with swath elevations because more snow is present in April than in June. The negative bias between the ALS from April and swath elevations could be directly attributed to the surface penetration of the radar in snow-covered areas. CryoVex ALS data were obtained with the goal of obtaining information about the penetration depth of the radar into the snowpack and not aligning it with processed CS2 data. In our study of assessing the performance of the swath processor, it would be optimal to attain ALS validation data from the summer months.

The main objective of swath processing is to survey ice-surface topography in higher spatial and temporal resolutions. The coastal regions of the ice sheets are rapidly changing, and the continued improvement of the observations of these regions can be advanced by methods such as swath processing.

**Author Contributions:** Conceptualization: N.H.A., L.S.S. and S.B.S.; methodology: N.H.A. and J.N.; software: N.H.A.; validation: N.H.A., M.W., J.N. and S.B.S.; formal analysis, N.H.A.; investigation: N.H.A.; data curation: N.H.A., J.N. and M.W.; writing—original-draft preparation: N.H.A.; writing—review and editing, N.H.A., L.S.S., S.B.S., M.W. and J.N.; visualization, N.H.A. All authors have read and agreed to the published version of the manuscript.

**Funding:** This research received no external funding.

**Institutional Review Board Statement:** Not applicable.

**Informed Consent Statement:** Not applicable.

**Data Availability Statement:** Cryosat-2 SARin data is freely available on ESA' homepage: https://science-pds.cryosat.esa.int/ (accessed on 2 June 2021), IceSat-2 ATL06 data is freely available on NSIDC: https://nsidc.org/data/atl06 (accessed on 2 June 2021), ArcticDEM 1 km resolution can be downloaded from https://data.pgc.umn.edu/elev/dem/setsm/ArcticDEM/ (accessed on 2 June 2021), OIB data is freely available on NSIDC: https://nsidc.org/data/icebridge/data_summaries.html (accessed on 2 June 2021), ALS over Austofnna is available at https://earth.esa.int/eogateway/search?text=cryovex (accessed on 2 June 2021) and the X-band data over Petermann is available upon request at DTU (ssim@space.dtu.dk).

**Acknowledgments:** We would like to acknowledge Brian Wegner from Fugro Geospatial Inc. for providing the Fugro airborne GEOSar radar data for the Petermann glacier as a part of logistical cooperation in Greenland. ArcticDEM was provided by the Polar Geospatial Center under NSF-OPP awards 1043681, 1559691, and 1542736. NASA's Operation IceBridge and ICESat-2 data were downloaded from NSIDC. Some data used in this study were acquired by ESA's CryoVEx campaigns.

**Conflicts of Interest:** The authors declare no conflict of interest.

## Abbreviations

The following abbreviations are used in this manuscript:

| | |
|---|---|
| ALS | Airbornelaser-scanner data |
| ATLAS | Advanced Topographic Laser Altimeter System |
| ATM | Airborne Topographic Mapper |
| CryoVEx | Cryosat Validation Experiment |
| CS2 | CryoSat-2 |
| DEM | Digital elevation model |
| DTU | Danish Technical University |
| ESA | European Space Agency |
| IS2 | ICESat-2 |
| LRM | Low-resolution mode |
| OIB | Operation ICEbridge |
| POCA | Point of closest approach |
| SAR | Synthetic aperture radar |
| SARIn | interferometric SAR |
| SIRAL | Synthetic Aperture Interferometric Radar Altimeter |
| STD | Standard deviation |

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
