# Peer review of "Regional Assessments of Surface Ice Elevations from Swath-Processed CryoSat-2 SARIn Data"

_remotesensing, doi:10.3390/rs13112213_

Round 1
Reviewer 1 Report
I would recommend improving the background in introduction and develop the literature review in this section. Also highlight the novelty of this work to the readers in introduction.

Author Response
We thank the reviewer for the comments in the pdf and have changed the manuscript accordingly.
We have improved the “background in intro and developed literature review” by extending the section from line 35 to 52 with additional references.
In Line 35 the following has been rephrased to “The method named Swath processing was developed using CS2 data in 2013 by \citep{Gray2013}, and…”
In line 52 the following has been added “The ongoing improvements of the swath processed data can additionally be seen with each new baseline where, among other things, the roll angle bias estimations and phase unwrapping are improved \cite{Garcia2021,Handbook_baselineD}.
We also emphasized the “novelty of this work” by noting the added value of investigating more than one coherence threshold compared to the operational products that uses one standard threshold.
In line 65 the following has been added “ Previous studies have shown great potential in using swath processing here we investigate the added value of regionally choosing different coherence thresholds – an aspect that has previously not been explored in the literature.”
Comments from pdf by reviewer 1
See the track changes file (diff.pdf) to see the actual corrections:
- Line 29: “. It is also here that the largest changes” Was changed to “,where the largest changes…”
- Line 32: Changed to “has been used to map ...” instead of “has mapped”
- Line 33: “The” has been added and “Its” has been removed
- Line 47: “Here” has been changed to “in this study”
- Line 57: “where” has been changed to “were”
- Line 59: “Consist” has been changed to “consists”
- Line 63: ”That” has been removed
- Line 65-69: Rewritten to one sentence instead of two:
“Through such detailed analysis of the regional performance of the swath processor we add to the current knowledge of how to improve the accuracy of surface elevations from swath processing. This will contribute to the overall goal of improving the ground coverage over these fast changing regions to minimize the possible underestimation of land ice volume changes observed from CS2 [23].”
Changed to:
“Through such detailed analysis of the regional performance of the swath processor we contribute to the overall goal of improving the ground coverage over these fast-changing regions to minimize the possible underestimation of land ice volume changes observed from CS2 [23].”
- Line 72: “, selected due to availability…” has been changed to ”. These regions were selected due to availability ...”
- Line 104:Comma added here: ”In this study, …”
- Line 110:”meters” has been changed to ”m”
- Line 126:Define Abbreviations ATLAS in parathesis “(Advanced Topographic Laser Altimeter System)” And also added to the Abbreviations list in line 370.
- Line 146:Citation changed to mention name of author here
- Line 156:Parenthesis removed
- Line 158:Has been changed from ”same date” to ”with same data”
- Line 214:”In the table, we” has be changed to ”The table reports”
- Table 2:Added degree symbol to 79N
- Line 229:”Table 2 shows that” has been removed
- Line 230:”As expected” has been removed
- Line 232:Reference to table 2 has been added here
- Figure 4:Remove highlighting of a and b
- Figure 5:Rewritten from ”In insets are shown the corresponding…”
to
”The corresponding histograms of the elevation differences are shown in the inserts”
- Line 328:”Investigate” has been change to ”Investigated” and “the influence on the crossover statistics by…” has been rephrased to “We investigated the results of the crossover statistics by… ”
- Line 337:Dot added at the end of sentence
- Line 338:”Find” has been change to ”found”
- Line 343:”Emphasize” has been change to ”emphasizes” and “importance of also using intra-mission” has been rewritten to “This emphasizes the significance of including the intra-mission…”
- Line345:”Attention on the season” has been changed to “The seasonality must be considered”
- Line 356:”This is done in order to enhance the measurements of the mass loss of the quickly changing coastal regions of the ice sheets.”
has been rewritten to
”The coastal regions of the ice sheets are changing at rapid speed and the continued improvement of the observations of these regions can be advanced by methods like swath processing.”
Reviewer 2 Report
The structure of this manuscript is professionally written, and the results also help related researchers understand the use of various materials. The author used CryoSat-2 SARIn data to evaluate ice surface elevations in the Arctic region, and used altimeter and radar data to verify. I think the author's work has an important contribution to the monitoring of sea ice melting in the Arctic, especially the selection of relevant thresholds is necessary to help the accuracy of the measurement. In the future, if researchers use CryoSat-2 SARIn data to analyze Arctic scientific issues, they can refer to this manuscript. I only have a several comment as follows:
- Figure 1: The author should mark the place names of the four research areas on the map, and the latitude and longitude should be marked on the map. Please add what is meant by the gray area surrounded by the coastline.
- Table 1: Please add more time details instead of just the month.
- Figure 3: Can the author explain in more detail the difference between the gray line segment and the red line segment? Why is there such a big change.
- Table 2: The "79N" and "Austfonna" areas are not marked in Figure 1 and Table 1. Where is the scope of these two areas?
Author Response
Reviewer 2 Comments and Suggestions for Authors:
The structure of this manuscript is professionally written, and the results also help related researchers understand the use of various materials. The author used CryoSat-2 SARIn data to evaluate ice surface elevations in the Arctic region, and used altimeter and radar data to verify. I think the author's work has an important contribution to the monitoring of sea ice melting in the Arctic, especially the selection of relevant thresholds is necessary to help the accuracy of the measurement. In the future, if researchers use CryoSat-2 SARIn data to analyze Arctic scientific issues, they can refer to this manuscript. I only have a several comment as follows:
Figure 1: The author should mark the place names of the four research areas on the map, and the latitude and longitude should be marked on the map. Please add what is meant by the gray area surrounded by the coastline.
We thank the reviewer for the suggestions and have made the changes accordingly.
To Figure 1 the area names have been added to the map instead of only letters, as well as the corresponding longitude and latitude for the maps.
The grey shaded data plotted as background for Greenland is the ArcticDEM with 1 km resolution. And this has now been added in the figure text with a reference to the data. See new figure below.
 Table 1: Please add more time details instead of just the month.
We agree that additional information could improve this table, however, we have approximately 15-25 satellite tracks pr. region over the time period of one month, which makes it almost daily revisits. Therefore, we have added the dates for the period where we have obtained data inside the given boundary box.
The following dates are added to table 1:
|
Helheim : |
9-24th of June 2019 |
|
79N: |
1- 30 th of April 2018 |
|
Austfonna: |
1-30 th of April 2016 |
|
Petermann: |
1-30 th of April 2014 |
Figure 3: Can the author explain in more detail the difference between the gray line segment and the red line segment? Why is there such a big change.
It is the unwrapping of the 2 pi phase jump seen in the original grey data. These need to be removed/corrected for in order to avoid misplaced echoes in the final elevation product. The red line indicates the corrected phase which is used for further processing.
See Sentence in Line 168-170 that has been changed from “An example of phase unwrapping is seen in Figure 3, where the original phase is shown in grey and the correctly unwrapped phase is shown in red.”
to
“An example of phase unwrapping is seen in Figure 3, where the original phase in grey shows a phase jump between range bin 400 and 500. The red line shows the correctly unwrapped phase which will be used for further processing”.
Table 2: The "79N" and "Austfonna" areas are not marked in Figure 1 and Table 1. Where is the scope of these two areas?
We hope that the addition of the names and coordinates to the four regions in Figure 1 makes this more clear.
They are marked with the red box in Figure 1, which indicates the extent of the validation data and hence the Boundary box used for crossover analysis showed in Table 2.
Reviewer 3 Report
Review of the manuscript ID remotesensing-1221282 entitled “Regional assessments of surface ice elevations from swath-processed CryoSat-2 SARIn data”
In the presented study, the Authors address the problem of improving the monitoring of ice cover changes of four different regions of the High-Arctic, based on the analysis of satellite data. They determined the ice cap surface altitude using CryoSat-2 SARIn data. In addition, the swath processing used in this study significantly increased the spatial coverage over conventional retracking of radar data, by using most of the information contained in the radar waveform to create a swath of elevation measurements. To evaluate the performance of the swath processor, the Authors performed validation of the processed data for data from all study areas. An accuracy was evaluated for both intra-mission cross-over elevation differences and comparisons with independent elevation data. Validation data included both airborne and spaceborne laser altimetry and airborne X-band radar data. The results obtained confirm the utility of the method used and highlight the importance of selecting an appropriate coherence threshold for swath processor performance. The conducted study also has a practical dimension and may contribute to better monitoring of the rate of cryosphere degradation and, consequently, to the elucidation of both the mechanism and effects of these changes.
In my opinion, the study itself as well as the obtained results and their interpretation do not raise any doubts. The layout of the paper is clear and the results was plausible in the validation process. The discussion is perhaps too little oriented towards referring to the results of other studies. So my only suggestion for improvement comes down to a suggestion to broaden the references in this part of the text. Other than these minor additions, I recommend acceptance of this manuscript for publication in the Remote Sensing Journal.
Author Response
Reviewer 3 Comments and Suggestions for Authors :
Review of the manuscript ID remotesensing-1221282 entitled “Regional assessments of surface ice elevations from swath-processed CryoSat-2 SARIn data”
In the presented study, the Authors address the problem of improving the monitoring of ice cover changes of four different regions of the High-Arctic, based on the analysis of satellite data.  They
determined the ice cap surface altitude using CryoSat-2 SARIn data. In addition, the swath processing used in this study significantly increased the spatial coverage over conventional retracking of radar data, by using most of the information contained in the radar waveform to create a swath of elevation measurements.  To evaluate the performance of the swath processor, the Authors performed validation of the processed data for data from all study areas. An accuracy was evaluated for both intra-mission cross-over elevation differences and comparisons with independent elevation data. Validation data included both airborne and spaceborne laser altimetry and airborne X-band radar data. The results obtained confirm the utility of the method used and highlight the importance of selecting an appropriate coherence threshold for swath processor performance. The conducted study also has a practical dimension and may contribute to better monitoring of the rate of cryosphere degradation and, consequently, to the elucidation of both the mechanism and effects of these changes.
In my opinion, the study itself as well as the obtained results and their interpretation do not raise any doubts. The layout of the paper is clear and the results was plausible in the validation process. The discussion is perhaps too little oriented towards referring to the results of other studies. So my only suggestion for improvement comes down to a suggestion to broaden the references in this part of the text.
Other than these minor additions, I recommend acceptance of this manuscript for publication in the Remote Sensing Journal.
We thank the reviewer for the suggestions and have added the following references to accommodate their suggestion.
In line 257 the following has been added “, which is also noted to be a challenging area in the study by \citet{Soerensen2018}”
In line 357 the following have been added “However, one should be aware of a decrease in radar penetration depth which can be caused by fluctuations in density due to summer melt events \cite{Otosaka2020}. “